# Identification of Copy Number Variations and Selection Signatures in Wannan Spotted Pigs by Whole Genome Sequencing Data: A Preliminary Study

**DOI:** 10.3390/ani14101419

**Published:** 2024-05-09

**Authors:** Wei Zhang, Yao Jiang, Zelan Ni, Mei Zhou, Linqing Liu, Xiaoyu Li, Shiguang Su, Chonglong Wang

**Affiliations:** 1Anhui Provincial Breeding Pig Genetic Evaluation Center, Key Laboratory of Pig Molecular Quantitative Genetics of Anhui Academy of Agricultural Sciences, Anhui Provincial Key Laboratory of Livestock and Poultry Product Safety Engineering, Institute of Animal Husbandry and Veterinary Medicine, Anhui Academy of Agricultural Sciences, Hefei 230031, China; zhangwei615527@126.com (W.Z.); 1229zhoumei@163.com (M.Z.); liulinqingllq@126.com (L.L.); lixiaoyudongke@163.com (X.L.); 2National Animal Husbandry Service, Beijing 100125, China; jiangyao133996@126.com; 3Anhui Provincial Livestock and Poultry Genetic Resources Conservation Center, Hefei 231283, China; nizelan@126.com

**Keywords:** Wannan spotted pig, Asian wild boar, copy number variation, selection signatures, whole genome resequencing

## Abstract

**Simple Summary:**

The aim of this study was to detect the copy number variations in 25 Wannan spotted pigs and 10 Asian wild boars with whole genome sequencing data. The selection signatures of Wannan spotted pigs compared to those of Asian wild boars were calculated with fixation index (Fst). A total of 195 selected CNVs were under selected, harboring 80 genes. The selected genes were associated with reproduction, fatty acid composition, immune system, ear size, and feed efficiency. This study provides a new insight for understanding the characteristics of Wannan spotted pigs based on copy number variations.

**Abstract:**

Copy number variation (CNV) is an important structural variation used to elucidate complex economic traits. In this study, we sequenced 25 Wannan spotted pigs (WSPs) to detect their CNVs and identify their selection signatures compared with those of 10 Asian wild boars. A total of 14,161 CNVs were detected in the WSPs, accounting for 0.72% of the porcine genome. The fixation index (Fst) was used to identify the selection signatures, and 195 CNVs with the top 1% of the Fst value were selected. Eighty genes were identified in the selected CNV regions. Functional GO and KEGG analyses revealed that the genes within these selected CNVs are associated with key traits such as reproduction (*GAL3ST1* and *SETD2*), fatty acid composition (*PRKG1*, *ACACA*, *ACSL3*, *UGT8*), immune system (*LYZ*), ear size (*WIF1*), and feed efficiency (*VIPR2*). The findings of this study contribute novel insights into the genetic CNVs underlying WSP characteristics and provide essential information for the protection and utilization of WSP populations.

## 1. Introduction

Pigs (*Sus scrofa domesticus*) are important agricultural animals that were domesticated during the early Neolithic period [1]. They serve several functions for human society: (1) Food supply: Pork is one of the most important meat products worldwide, providing high levels of protein, vitamins, and minerals to meet people’s nutritional needs; (2) Economic value: Pig farming is a large industry that provides considerable economic benefits to a country; (3) Agricultural recycling: They can help dispose of organic waste and leftover matter in farmlands, convert it into organic fertilizer, and keep farmlands fertile, reducing waste disposal and promoting sustainable agriculture; (4) Experimental animals: Their anatomy and physiology are similar to those of humans and therefore, they can be used to study human health problems, for drug testing, and in organ transplantation; and (5) Culture: In traditional Chinese culture, pigs are regarded as symbols of wealth, prosperity, and happiness. Pigs also play an important role in festivals and celebrations, such as the Chinese Spring Festival. In general, a complete understanding of the germplasm characteristics of pig breeds is key to the conservation and utilization of pig breed resources.

The pig reference genome assembled in 2012 was of landmark significance [2]. With advances in science and technology, a highly contiguous genome assembly (Sscrofa11.1) was accomplished [3], which led to the discovery of genomic variations with greater accuracy. Numerous studies have been performed to identify single nucleotide polymorphisms (SNPs) and elucidate their relationship with important economic traits [4,5,6]. In addition to revealing the effects of SNP on phenotypes, another form of genomic variation, known as copy number variation (CNV), has not been extensively studied. CNV is defined as a fragment ranging from 50 bp to several Mb and can be classified as a duplication (Dup) or deletion (Del) [7]. Compared with SNPs, CNVS have significant genomic effects, such as direct effects on gene dosage and indirect changes in gene expression [8]. Research has shown that 25% of the identified CNVs do not have linkage disequilibrium with the detected SNP, indicating that CNVs play an irreplaceable role compared with SNPs [9]. Many studies have been conducted to determine the relationship between CNVs and important economic traits in humans and animals. For instance, in humans, *AMY1* is associated with the digestion of starch [10], and *MKL1* explains the adaptation to the plateau environment in Tibetan humans [11]. In *Bos taurus*, *TSPY* is important for male embryonic development [12], and *SYT11* is positively related to growth conformation traits [13]. In sheep, duplication of *ASIP* results in differences in coat color [14]. In pigs, CNV in *MSRB3* could increase porcine ear size [15], and CNV in *PELP1* could explain the differences in intramuscular fat content [16]. These studies show that CNVs play an irreplaceable role in revealing complex traits. Considering the huge number of breed resources worldwide, it is vital to detect CNVs and elucidate the genetic basis of excellent germplasm characteristics.

The Wannan spotted pig (WSP) is native to the Kecun area, Anhui, China, located across 29°24′–30°11′ N and 117°38′–118°53′ E. This breed has a history of at least 500 years and is recorded in “annals of Xin’an” by Yuan Luo and “annals of Huizhou Prefecture” by Hongzhi Ming from the Song Dynasty. WSPs are deeply loved by the local people, not only for providing meat protein and increasing farmers’ income but also for their position in local culture and history. With a long history of breeding, the WSP has developed genetic characteristics of rough feeding resistance, disease resistance, and excellent meat quality and has been recorded in the “China National Commission of Animal Genetic Resource” [17]. According to the third National Survey of Livestock and Poultry Genetic Resources in 2021, 8558 boars (85 boars; 2311 sows) will be stored in Anhui Province. However, research on WSPs at the genome level is relatively scarce. Our previous studies preliminarily elucidated the characteristics of WSPs based on RNA-seq and whole-genome resequencing (mainly SNP) [18,19]. Considering the critical role of CNV in economic traits, the detection and functional analysis of CNV in WSP populations are needed.

Subsequently, the main objectives of this study were to (1) detect CNV in WSP and AWB populations; (2) identify the selection signatures of the WSP compared to the Asian wild boar (AWB) based on CNV by calculating the Fst value; and (3) perform functional genomic annotation of the selected CNV. Our findings can increase understanding of WSPs and provide guidance for future protection and breeding.

## 2. Materials and Methods

### 2.1. Ethics Statement

This study was conducted in accordance with and was approved by the Animal Care Committee of the Anhui Academy of Agricultural Sciences (Hefei, China; no. AAAS2020-04).

### 2.2. Sample Collection and Sequencing Data Processing

The blood samples from twenty-five unrelated WSPs (♀10, ♂15; Figure 1) were collected from a provincial conversation farm in Huangshan city, Anhui province, China. Blood was collected from the auricular vein by using a blood collection needle and stored in an anticoagulation tube. Genomic DNA was extracted using the standard phenol–chloroform method [20] and assessed using a 0.5% agarose gel and Nanodrop spectrophotometer (Thermo Fisher Scientific, Waltham, MA, USA). Library construction and sequencing went as follows: (1) DNA fragmentation; (2) purification of the target fragments; (3) addition of adapter ligation; and (4) PCR amplification and sequencing. Sequencing was performed on the Illumina NovaSeq 6000 platform (Illumina, San Diego, CA, USA) with paired-end 150 bp reads using the Novogene service (Beijing, China). The sequencing reads were processed with NGSQCToolkit [21], including removing reads containing adapter or poly-N and low-quality reads with >30% base having Phred quality ≤25.

To assess the selection signatures of WSPs compared to Asian wild boars (AWBs), ten sequenced AWB samples were combined. Six accessions were obtained from our previous study, with accession numbers SRR13630747–SRR13630752 [22]. The other four were downloaded from the NCBI database (https://www.ncbi.nlm.nih.gov/, (accessed on 15 March 2024)) with accession numbers ERR173220, ERR173222, SRR652378, and SRR652379 [2,4].

### 2.3. Detection of CNVs in WSP and AWB Population

The clean reads were mapped to the pig reference genome 11.1 (https://www.ncbi.nlm.nih.gov/datasets/genome/GCF_000003025.6/ accessed on 15 March 2024) using the Burrows–Wheeler Aligner with default parameters [23]. Population CNVs on autosomes were detected by combining the Manta v.1.6.0 [24] and Paragraph v2.4a software [25]. The concrete procedures were: (1) detection of CNVs in each sample with Manta; (2) merging the CNVs based on the CNV type and removing redundancy on the basis of genomic location and CNV length; (3) genotyping the CNVs in each sample by Paragraph based on the results with removed redundance; (4) removing the redundancy at the population level based on location information (del is overlapping by 50%; dup are set to overlap 90%) and genotyping results (population typing consistency ≥ 0.95); and (5) the quality controls were as follows: ABS(INFO/SVLEN) ≤ 10,000,000, INFO/ExcHet ≥ 0.05, F_MISSING ≤ 0.2, and INFO/MAF > 0. PLINK software v.1.90 [26] and R (v4.2.0) were used to calculate the frequencies of the two populations and for separate visualization.

### 2.4. Identification of Selection Signatures

We explored and genotyped bi-allelic CNVs in WSPs and AWBs in light of population genetics by Manta and Paragraph. These genotypes were used to calculate the allele frequency of each CNV locus. The genetic differentiation of the two populations based on CNVs was calculated using the “-weir-fst-pop” in VCFtools [27] with the fixation index (Fst) method [28], which is a widely used statistic in CNV studies [29,30]. The value of Fst ranged from 0 to 1. A larger value indicated greater differentiation at the CNVs; otherwise, a smaller variation was observed. The formula for the Fst calculation was Fst = (Ht − Hs)/Ht, where Ht is the expected heterozygosity of the population and Hs is the expected heterozygosity of the subgroup. Fst was calculated for 19,537 bi-allelic CNVs. In this study, a CNV with an Fst value in the top 1% of all Fst values was regarded as a selected CNV. All selected CNVs were aligned to *Sus scrofa* 11.1 to obtain the genes harboring CNVs. Gene Ontology (GO) and Kyoto Encyclopedia of Genes and Genomes (KEGG) analyses were performed to annotate selected genes using KOBAS (http://kobas.cbi.pku.edu.cn/ accessed on 15 March 2024). The terms and pathways exhibiting *p*-values < 0.05 were considered significant.

## 3. Results

### 3.1. Detection of CNVs in WSP and AWB

The clean data of 25 WSPs and 10 AWBs were 724.83 Gb and 253.7 Gb, respectively. The average depth of the WSPs was 10.11× and ranged from 9.11× to 12.82× (Appendix A). The average depth of the AWBs was 9.73× and ranged from 6.24× to 12.53× (Appendix A). A total of 978.53 Gb was used for CNV detection and selection signature analyses. A total of 14,161 CNVs, covering ~18.12 Mb (0.72% of the pig genome), were detected in the 25 WSP, consisting of 13,671 deletions and 490 duplications (Table 1). In AWBs, 14,355 CNVs, covering ~22.10 Mb (0.88% of the pig genome), were detected, which consisted of 13,919 deletions and 436 duplications (Table 1). Detailed information on the CNVs in WSPs and AWBs is provided in Appendix A. The number of CNVs on each chromosome is consistent with its length. In the WSP, chromosome 1 had the largest number of CNVs (1332), and chromosome 18 had the lowest (414) (Appendix A). The numbers of Del and Dup genes on each chromosome are shown in Figure 2A. In AWB, chromosomes 1 and 18 had 1414 and 439 CNVs, respectively (Appendix A). The numbers of Del and Dup genes on each chromosome are shown in Figure 2B.

After merging the CNVs of the WSP and AWB based on the procedures in Materials and Methods, 19,537 CNVs (18,923 Del and 614 Dup; Appendix A) were obtained, covering 22.91 Mb in length (approximately 1.2% of the pig genome). To assess the distribution of CNVs in gene-related regions, the merged CNVs were annotated and revealed that the intergenic region has the largest number of CNVs (8923), following the intronic, ncRNA, exonic, and 3′UTR regions (Table 2). To assess the frequency of CNVs in the two populations, they were divided into ten groups (0–0.1, 0.9–1, see Appendix A; Figure 3A,B). In WSPs, the 0–0.l group had 7357 Dels, the largest of the ten groups, covering 38.88%. The CNV trends in the two populations were similar.

### 3.2. Patterns of Selection Signatures

The selection signatures in the autosomes of the WSP were calculated using Fst. The threshold for the top 1% of the total FST values was 0.8293. A CNV with an Fst value greater than 0.8293 was selected. A total of 195 CNVs were identified (Appendix A). A Manhattan plot of the FST statistics is shown in Figure 4. After annotating the selected CNVs in the pig genome, 80 genes were identified in the selected CNVs (Appendix A). Figure 5 shows the results of the functional annotation analysis of the selected genes. A total of 32 GO terms were enriched at level 2 GO enrichment (Appendix A), which consisted of immune system processes (GO:0002376, 10 genes), regulation of biological processes (GO:0050789, 52 genes), responses to stimuli (GO:0050896, 34 genes), reproductive processes (GO:0022414, 2 genes), reproduction (GO:0000003, 2 genes), and metabolic processes (GO:0008152, 45 genes). Twelve pathways were enriched in the KEGG analyses (Appendix A), including fatty acid biosynthesis (KO00061, 2 genes), metabolic pathways (KO01100, 17 genes), biotin metabolism (KO00780, 1 gene), and fatty acid metabolism (KO01212, 2 genes).

## 4. Discussion

WSPs play an important role in the protection of genetic resources and local characteristics and the development of local agriculture and cultural inheritance. (1) Genetic resource protection: WSPs are one of the excellent resources in China, which is very important for the protection and inheritance of this native genetic resource; (2) Local characteristics: the WSP is a local, rural, characteristic breed, representing the culture and tradition of the southern Anhui region; (3) Strong adaptability: WSPs have adapted to the local climate environment and feeding conditions and are a common breed of local farmers; (4) Excellent meat quality: the meat of WSP is fresh and tender, is juicy, and has a good taste, which is favored by consumers; (5) Economic benefits: The breeding of WSPs can bring economic benefits to local farmers, improve livelihoods, and promote local economic development.

In this study, we sequenced 25 WSPs to identify CNVs and genes. A total of 724.83 Gb of data were generated for the WSP, and 14,161 CNVs were detected. For the signatures of the selection analysis, 10 AWB were combined. A total of 195 CNVs harboring 80 genes were found to be under selection. Functional analysis revealed that these genes are associated with important traits.

Elucidating the genetic basis of pig reproductive performance is the key to optimizing offspring production. Indigenous Chinese pig breeds are characterized by high fertility rates. In this study, we found that several genes in the selected CNV regions were associated with reproduction. Galactose-3-O-sulfotransferase 1 (*GAL3ST1*), also known as cerebroside sulfotransferase (CST), is involved in spermatogenesis [31]. The *GAL3ST1* was found selected in (dwarf surf clam) and associated with male reproduction [32]. By observing GAL3ST1 knockout mice, it was found that males displayed sterility, which resulted from a block in spermatogenesis before the first meiotic division [33]. A genome-wide association analysis of bulls showed that *GAL3ST1* was significantly associated with sperm concentration [34]. Using the GeneSeek Genomic Profiler Porcine HD BeadChip (Neogen Corporation, Lansing, MI, USA), *GAL3ST1* was found to be associated with spermatogenesis in 223 pigs [35]. *GAL3ST1* is found on chromosome 14 in pigs. In the present study, a DUP (61 bp, start = 47,474,122 bp, end = 47,474,183 bp) was found in the intronic region. SET domain-containing 2 histone lysine methyltransferase (*SETD2*), an H3K36me3 methyltransferase, participates in the maintenance of chromatin architecture, transcription elongation, genome stability, and other biological events. *SETD2* was first found as selected. Recent research on *SETD2* in mice revealed that *SETD2* deficiency results in a series of alterations in the oocyte epigenome, such as the loss of H3K36me3 and failure to establish the correct DNA methylome. More importantly, depletion leads to defects in oocyte maturation and subsequent one-cell arrest after fertilization [36]. In mice, deletion of *SETD2* results in developmental delay in early embryos [37,38]. In a study on the effects of *SETD2* in zebrafish, SETD2-null zebrafish were fertile. However, they have a small body size due to insufficient energy metabolism and protein synthesis [39]. In the present study, Del (114 bp, start = 29,915,687, end = 29,915,801) was detected in the intronic region. Considering the above findings, *GAL3ST1* and *SETD2* could be regarded as candidate genes that regulate reproductive traits in pigs. Further functional studies are required to confirm this hypothesis.

Meat quality is a key factor in the sustainable development of the pork industry. Fatty acids, which are important flavor precursors, directly affect the sense of smell and taste in meat sensory reactions and are also one of the important factors leading to differences in the taste of different pork types, indirectly affecting their acceptability to consumers and the value of meat. In this study, several genes (*PRKG1*, *ACACA,* and *ACSL3*) were found to be associated with fatty acid metabolism. cGMP-dependent kinase 1 (*PRKG1*) regulates lipolysis in adipocytes to release fatty acids and glycerol via the hydrolysis of triacylglycerol. The *PRKG1* was found as selected in the pigs of our previous study [19]. In cattle, post-GWAS identified that *PRKG1* has a positive effect on milk fatty acids, especially medium-chain saturated fatty acid traits [40]. Moreover, *PRKG1* knockout mice have decreased triglyceride stores in brown adipose tissue [41]. In pigs, a GWAS on fatty acid composition in 691 Ningxiang pigs revealed that *PRKG1* was associated with saturated fatty acid traits [42]. According to RNA-Seq analysis, *PRKG1* exhibits differential expression between high and low fatty acid composition groups in the muscle [43]. Acetyl-CoA carboxylase alpha (*ACACA*), an important enzyme in lipid metabolism, controls de novo fatty acid biosynthesis [44]. The ACACA was found selected in pigs in a previous study [45]. Several studies have demonstrated that *ACACA* has a significant impact on the fatty acid profiles of cattle, including C10:0, C14:0, and C13:0 [46,47,48,49]. In a transcriptomic and lipid metabolomic analysis of beef cattle, *ACACA* positively correlated with total MUFA [50]. In the Puławska breed (a Poland native pig breed originating from the Lublin region), *ACACA* exhibited a relationship with IMF content, with differences reaching 20% [51]. Acyl-CoA synthetase long-chain family member 3 (*ACSL3*) mediates the synthesis of diacylglycerol, affects the secretion of very low density lipoproteins, and stimulates fatty acid oxidation and lipid accumulation in mammals [52,53]. The *ACSL3* was first found as selected in pig. Previous studies have shown that *ACSL3* knockdown significantly reduces the activity of lipid-producing transcription factors and regulates fat production in the liver [54]. Inhibition of intestinal *ACSL3* reduces lipid synthesis [55]. In cattle, knockdown of *ACSL3* expression leads to a decrease in lipid content in cattle adipocytes [56]. In pigs, *ACSL3* overexpression and knockdown promote the accumulation of lipid droplets in intramuscular preadipocytes [57]. Bile acids play crucial roles in controlling lipid and glucose metabolism and directly regulating the fatty acid pathway [58]. Previous research revealed that UDP glycosyltransferase 8 (*UGT8*) participates in bile acid signaling. Moreover, in a study on Nanchukmacdon pigs, *UGT8* expression was positively correlated with meat quality [59].

Furthermore, we identified genes related to other important traits. The immune system is important for pig health and is a key factor in the pig industry. Lysozyme (*LYZ*), an antimicrobial enzyme, is primarily recognized as an important component of the innate immune system [60]. The LYZ was found as selected in pigs in a previous study [61]. In buffaloes, lysozyme expression was positively associated with antibacterial activity in the buffalo mammary glands [62]. In mice, LYC protects the intestinal epithelium from oxidative injury induced by DON exposure [63]. In piglets, the addition of lysozyme to dietary supplements can improve the development and function of the intestine and protect against enterotoxigenic *Escherichia coli* infection [64,65]. Ear size is an obvious trait used to distinguish between pig breeds. WNT inhibitory factor 1 (*WIF1*) could inhibit activity of the Wnt/β-catenin pathway that can regulate proliferation and differentiation in many tissues, such as controlling the growth of connective tissue by regulating the connective tissue growth factor [66]. The *WIF1* was found as selected in pigs in our previous study [29]. Based on an association analysis, *WIF1* was found to regulate the ear size of pigs and dogs [29,67]. Feed efficiency is one of the main factors determining production costs in the pig industry. Vasoactive intestinal peptide receptor 2 (*VIPR2*), a transmembrane glycoprotein, was selected in this study. The *VIPR2* was found as selected in pigs in our previous study [30]. In humans, *VIPR2* plays an important role in controlling fat deposition [68]; in mice, *VIPR2*-knockout results in growth inhibition, decreased fat mass, and increased lean mass [69]; and in pigs, *VIPR2* is associated with feeding efficiency [70].

In this study, we detected CNVs in WSPs for comparison with AWBs. Genes associated with important traits were also identified. However, this study has some limitations. First, the sample size of each population was insufficient to represent the total population. Second, because of African swine fever, we could not collect phenotypes and samples to verify the function of CNVs. Nevertheless, to some extent, this study supports the understanding of CNVs in WSPs and elucidates the effects on domestication.

## 5. Conclusions

We conducted a comprehensive detection of CNVs in WSPs at the genomic level, comparing them with wild pigs to pinpoint selectively favored CNVs. These identified CNVs are linked to key economic traits, thereby enhancing the genetic variation landscape of the WSP. Our findings create a robust basis for future breeding efforts aimed at trait enhancement and the preservation of genetic diversity in WSPs. Moreover, this study offers substantial backing for the analysis of significant economic traits in local pigs through the lens of CNVs, potentially speeding up the breeding process and boosting the competitive edge of local pig breeds in the marketplace.

## Figures and Tables

**Figure 1 animals-14-01419-f001:**
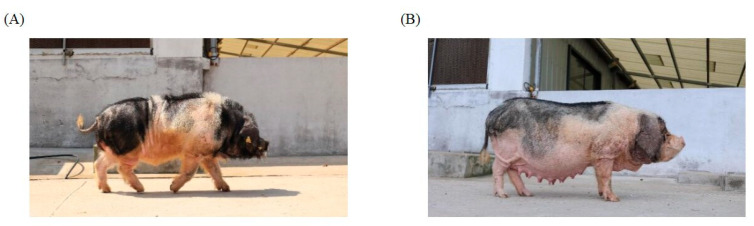
Pictures of Wannan spotted pig: (**A**) male; (**B**) female.

**Figure 2 animals-14-01419-f002:**
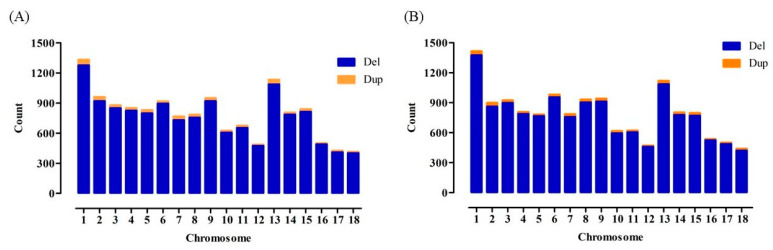
Numbers of CNVs identified across autosomes: (**A**) the WSP; (**B**) the AWB.

**Figure 3 animals-14-01419-f003:**
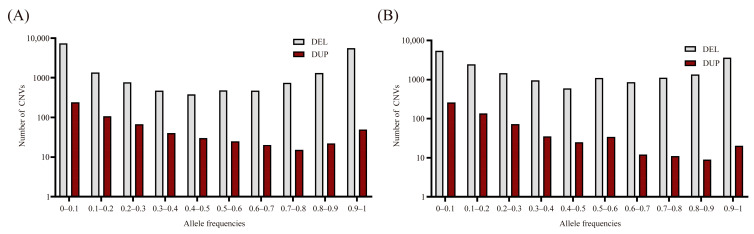
The allele frequencies of CNVs in the WSP (**A**) and AWB (**B**).

**Figure 4 animals-14-01419-f004:**
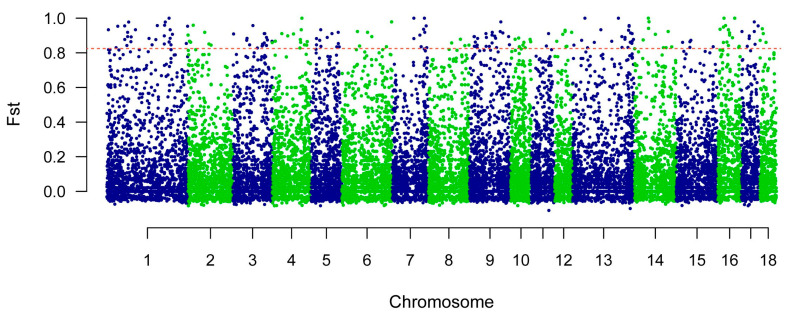
The Manhattan plot of the Fst based on CNVs, the green/blue dots represent the Fst value of all CNVs, the red dashed line represent the threshold line of top 1% of Fst.

**Figure 5 animals-14-01419-f005:**
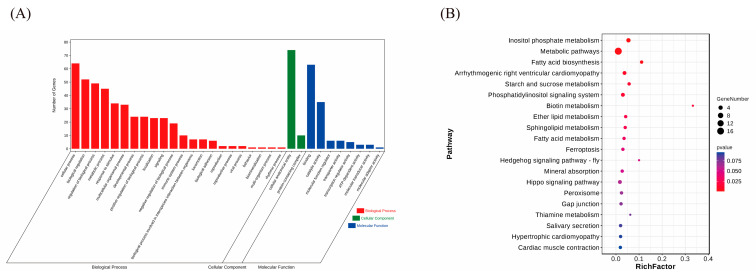
(**A**) GO analysis of the selected genes, red referring to biological process, green referring to cellular component, and blue referring to molecular function. (**B**) KEGG analysis of the selected genes, the size of dot referring to the number genes related to pathway, and the red-to-blue gradient indicating the significance of the *p*-value change.

**Table 1 animals-14-01419-t001:** The number and length of CNV in WSP and AWB.

Breed	Total	Number of Variants		Total Length (bp)/
	Number	Del	Dup	Genome Ratio
WSP	14,161	13,671	490	18,121, 370/0.72%
AWB	14,355	13,919	436	22,097, 264/0.88%

**Table 2 animals-14-01419-t002:** Annotation of the merged CNVs.

Classification	No. of Variants
Downstream	161
Upstream	110
Upstream; downstream	3
Exonic	205
Intronic	8570
Intergenic	8932
ncRNA	1307
Splicing	14
UTR3	188
UTR5	47

## Data Availability

The raw data supporting the conclusions of this article will be made available by the authors, without undue reservation.

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
