# Peer review of "Identification of Copy Number Variations and Selection Signatures in Wannan Spotted Pigs by Whole Genome Sequencing Data: A Preliminary Study"

_animals, 2024, doi:10.3390/ani14101419_

Round 1

Reviewer 1 Report

Comments and Suggestions for Authors

In this manuscript, the authors have attempted to identification of Copy Number Variations and Selection Signa- 2 tures in Wannan Spotted Pigs by Whole Genome Sequencing 3 Data.

The article has few minor deficiencies:

The research is very interesting, but as the Authors emphasise, the population is very small, which makes it impossible to define the conclusions clearly. I therefore recommend that the title of the manuscript  be changed to state that this is preliminary/pilot study.

Section Materials and Methods lines 93-111: Blood collection is a procedure that causes pain and stress to the animals and therefore the collection of such samples should be done with the approval of the Ethics Committee. The Authors should add in this section information on relevant regulations or Ethics Committee approval.

Section Materials and Methods lines 120: I suggest to the Authors to change redundance to redundancy

Section Results line 154: The Authors titled Tables 1 „The statistic of CNV in WBP and AWB”. This description seems inadequate especially as there is no subsection on statistical description in the Materials and methods section.

Section Discussion lines 227-228: The Authors should include here which exact genes.

Section Discussion Line 240: The reader may not know which breed this refers to. It would therefore be appropriate to specify that this information refers to a breed of pig.

I recommend the manuscript to be considered for publication after minor revision.

Author Response

Dear reviewer 1:

Thank you for your comments concerning our manuscript entitled “Identification of Copy Number Variations and Selection Signatures in Wannan Spotted Pigs by Whole Genome Sequencing Data” (ID: animals-2976933). Those comments are all valuable and very helpful for revising and improving our paper, as well as the important guiding significance to our researches. We have studied comments carefully and have made correction which we hope meet with approval. Revised portion are marked in red in the paper. The main corrections in the paper and the responds to the comments are as flowing:

  1. The research is very interesting, but as the Authors emphasize, the population is very small, which makes it impossible to define the conclusions clearly. I therefore recommend that the title of the manuscript be changed to state that this is preliminary/pilot study.

Response: It is really true as Reviewer suggested that the title of the manuscript should state this is a preliminary study. We have changed the title as “Identification of Copy Number Variations and Selection Signatures in Wannan Spotted Pigs by Whole Genome Sequencing Data: A preliminary study”, and marked with red in line 2-4.

  1. Section Materials and Methods lines 93-111: Blood collection is a procedure that causes pain and stress to the animals and therefore the collection of such samples should be done with the approval of the Ethics Committee. The Authors should add in this section information on relevant regulations or Ethics Committee approval.

Response: We are very sorry for our negligence of this mistake. This is very important that provide Ethics Committee approval in  Section Materials and Methods. We have added the Ethics statement in line 94-97 with red.

  1. Section Materials and Methods lines 120: I suggest to the Authors to change redundance to redundancy

Response: Thank you for your valuable suggestions. We have changed the redundance to redundancy and marked with red in line 124.

  1. Section Results line 154: The Authors titled Tables 1 „The statistic of CNV in WBP and AWB”. This description seems inadequate especially as there is no subsection on statistical description in the Materials and methods section.

Response: It is really true as Reviewer suggested that the title of Tables 1 is inadequate. We have revised the title as “The number and length of CNV in WBP and AWB” and marked with red in line 159.

  1. Section Discussion lines 227-228: The Authors should include here which exact genes.

Response: Thank you for your constructive suggestions. It is quite important providing the exact genes in this sentence for better understanding. We have added the exact genes and marked with red in line 234.

  1. Section Discussion Line 240: The reader may not know which breed this refers to. It would therefore be appropriate to specify that this information refers to a breed of pig.

Response: Thank you for your valuable suggestions. We are sorry for this mistake. It is quite important for readers to understand what the author want to express. We have added the essential information about the Puławska breed and marked with red in line 248-249.

Reviewer 2 Report

Comments and Suggestions for Authors

The paper is dedicated to elucidation of CNV patterns and selective sweeps in the genome of Wannan Spotted Pigs in comparison with Asian wild boars. The Introduction is nicely and interesting written. The revelation of new variants underlying the phenotypic, adaption, and economic important traits in genomes of local pigs is relevant to preserve and to ensure their sustainable management for future generations. The photographs of WSP sow and WSP boar are useful for attracting potential readers attention. 

However, there are several unclear points. 

1. The major question is appropriacy of the approach employed for searching selection signatures based on CNVs. Could you please provide brief explanation and justification (based on the published experience of other researchers) in the Materials and Methods why this approach was applied? The CNV patterns are different for domestic WSP and wild boars. 

2. L156-157 What do you mean by merging the WSP and AWB CNVs?

3. Could you please clarify why Asian wild boars (not other Chinese local or commercial pig breeds) were used as comparison group? 

4.  The CNVs were not validated by qPCR. Why? 

5.  The threshold is missing in Figure 4. Please male relevant corrections 

6. Were the identified genetic variants novel or were reported previosly for Sus scrofa? Please mention in Duscussion and Conclusion. 

 7. Please check that all gene names are in italics.

8. L88 Please decipher «AWB» as the first mentioned in the main text. 

Comments on the Quality of English Language

The English is fine 

Author Response

Dear Reviewer 2:

Thank you for your comments concerning our manuscript entitled “Identification of Copy Number Variations and Selection Signatures in Wannan Spotted Pigs by Whole Genome Sequencing Data” (ID: animals-2976933). Those comments are all valuable and very helpful for revising and improving our paper, as well as the important guiding significance to our researches. We have studied comments carefully and have made correction which we hope meet with approval. Revised portion are marked in red in the paper. The main corrections in the paper and the responds to the comments are as flowing:

  1. The major question is appropriacy of the approach employed for searching selection signatures based on CNVs. Could you please provide brief explanation and justification (based on the published experience of other researchers) in the Materials and Methods why this approach was applied? The CNV patterns are different for domestic WSP and wild boars. 

Response: Thank you for your valuable suggestions. We have added “The FST were widely used to detect the selection signatures in pig” and references based on the published experience. They were marked with red in line 132-133.

  1. L156-157 What do you mean by merging the WSP and AWB CNVs?

Response: Thank you for your constructive suggestions. We are sorry for vague expressions. We have revised as follows “After merging the CNVs of the WSP and AWB based on the procedures in materials and methods” and marked with red in line 161-162.

  1. Could you please clarify why Asian wild boars (not other Chinese local or commercial pig breeds) were used as comparison group? 

Response: Thank you very much for your valuable comment. When we carried out CNV detection and selection signal analysis of Wannan spotted pigs, the main basis for selecting Asian wild pigs at that time was: local pigs in China were domesticated from Asian wild pigs, so we wanted to look at the CNV situation between them and the detection of CNV selection signals, in order to identify some candidate genes related to important economic traits of Wannan spotted pigs. It is also of great significance to select commercial pig breeds or other local Chinese pig breeds as reference groups. In the future, we will select commercial pig breeds or local pigs as reference to detect CNV and select signals, with a view to elaborating the germplasm characteristics of Wannan spotted pigs more specifically.

  1. The CNVs were not validated by qPCR. Why? 

Response: It is true as reviewer suggested. Due to the situation of the African swine fever, we would collect the tissues to validate the CNVs in good conditions.

  1. The threshold is missing in Figure 4. Please male relevant corrections 

Response: Thank you very much for your advice. We are sorry for this mistake. We have corrected and the new version has threshold line.

  1. Were the identified genetic variants novel or were reported previously for Sus scrofa? Please mention in Discussion and Conclusion. 

Response: Thank you very much for your suggestive advice. We have mentioned in the discussion part with line208-209, 219,236-237, 245,253,266, 275,279.

  1. Please check that all gene names are in italics.

Response: Thank you very much. We have checked all the gene names in manuscript and revised with italics marking with red.

  1. L88 Please decipher «AWB» as the first mentioned in the main text. 

Response: We are very sorry for our negligence of this mistake. We have deciphered «AWB» as the first mentioned in line 88-89 and marked with red.

Round 2

Reviewer 2 Report

Comments and Suggestions for Authors

In general, the authors addressed the raised comments. 

L132-133 The comment was not about the using of method based on Fst calculation for detection the selection signatures in pigs or other livestock species. Could you provide cases of calculation Fst based on CNV patterns, which are different for domestic and wild species?  

Author Response

Dear reviewer 2:

Thank you for your comment concerning our manuscript entitled “Identification of Copy Number Variations and Selection Signatures in Wannan Spotted Pigs by Whole Genome Sequencing Data: A preliminary study” (ID: animals-2976933). The comment is valuable and very helpful for revising and improving our paper, as well as the important guiding significance to our researches. The respond to the comment is as flowing:

L132-133 The comment was not about the using of method based on Fst calculation for detection the selection signatures in pigs or other livestock species. Could you provide cases of calculation Fst based on CNV patterns, which are different for domestic and wild species?  

Response: Thank you for your valuable suggestions. It is true as suggested that the CNV patterns are different for domestic and wild pig. So, in this study, we first call the CNV with individual level. Secondly, merging the CNVs based on the CNV type and removing redundancy on the basis of genomic location and CNV length. Then merging the CNVs of Wannan spotted pigs and Asian wild boars. The calculation Fst were based on the merged CNV that could make sure the patterns are accordance.